# Cultural Production, Reproduction and Subversion of Gender Stereotyping among Pre-Service Science Teachers: Insights from Science Educators

Daniel Allu *[iD], Nadaraj Govender * and Angela James *

School of Education Science and Technology Education, University of KwaZulu-Natal, Durban 4041, South Africa
* Correspondence: asilika68@gmail.com (D.A.); govendern37@ukzn.ac.za (N.G.); jamesa1@ukzn.ac.za (A.J.)

**Abstract:** We perceive contestations between science, education and women's engagement and have raised disruptions in their act of knowing and mobility in science education. This study explored science educators' views, beliefs and actions of reproduction and subversion of gender stereotyping at a teacher education college in Nigeria. Six science educators were selected based on comprehensive gender information that facilitated conduction of the study. The six educators were purposively selected out of 11 educators who completed and returned the questionnaire. A qualitative approach and case study framed the research using instruments such as questionnaires, interviews, and classroom observation schedules. Thematic analysis and coding were done. Educators consciously and unconsciously reproduced gender stereotypes beliefs and practices. Educators explicitly and implicitly engaged in unequal distribution of cognitive activities amongst pre-service teachers influenced by their practice of cultural norms and patriarchal ideology. The findings revealed multiple oppressions females faced, contradictory science classrooms, and political and democratic classroom space for negotiating and renegotiating discriminatory classroom beliefs, perception and views of educators during science engagements. However, several possibilities such as political advocacy, productive activism and transformative resistance for educators to re-negotiate discriminatory gendered space through constructive gender equality awareness for freedom and intellectual growth in science education could be emancipatory possibilities.

**Keywords:** gender stereotype; production; reproduction; subversion; pre-service teachers; counter hegemony; oppression; activism; freedom and transformation

## 1. Introduction

In recent years, gender discourse and the construct equality has been included in policy documents and curricula in many countries. This might have informed feminist activism and politics that rekindle new gender insights into education. Nevertheless, gender and gender inequality in Science, Technology, Engineering and Mathematics (STEM) has been under attack in Europe, USA and in Africa which suggest there is crisis in feminist teaching models and awareness [1]. It is our concern that mainstreaming gender pedagogy in science, technology, engineering and mathematics would support science educators and science students with tools to critique and problematize ideas, hence challenging the epistemological foundations of knowledge construction. This could enhance students' critical thinking, creativity and conceptual understanding in science. No subject, content and curriculum will continue to remain the same when viewed through the lens of gender thinking. Therefore, it is vital to explore the content, context and discriminatory practices of science, technology, engineering and mathematics in higher education where gender and feminist approaches, skills and pedagogies are present and absent. This is in concert with Morris et al. [1] who argue that most leaders, policymakers and educators, insulate from knowledge produced through a feminist lens. However, academia has witnessed novel scholarships on cultural and gender studies and how this impacts gender discriminatory

practices in science education [2,3]. The advocacy for increased engagement of women in Science, Technology, Engineering, and Mathematics (STEM) fields extends locally and globally. For example, United Nations' agenda and Goal 5 on gender equality, caution that many countries are not adequately utilizing the potential of women and girls. Research in science, technology, engineering, and mathematics indicates that, although the population of women is about half of the world's total, only 30% of women are involved in scientific research [4]. Though women outnumber men in agricultural output, men outnumber females either as research students, science educators, researchers, or workers in STEM fields [5].

This is in line with Organization for Women in Science and Developing World-(OWSD) which comments that disparities are evident in workplaces and the level of responsibilities accorded to two genders differs. For instance, men get better pay and are concentrated at the upper management level with leadership opportunities and decision-making powers which they use to discriminate against and dominate women [6,7].

However, in Nigeria, as compared to South Africa, the United States of America, Singapore, Sweden and Europe, gender research in science education is limited. Gender research in Nigeria mostly focuses on inequality and gender discriminatory practices between male and female learners in primary and high school, university and college science modules, with a focus on quantitative data [8,9]. The research reveals a lack of participation, lack of enrolment, low performance, and poor attitude of females to study science. The gender research approach has re-echoed minimal interrogation of entrenched historical, cultural, and political power relations and patriarchy that reinforce inequity and discrimination due to superficial anti-gender politics and critical research for change and transformation. Further, in Africa and globally, little is said about gender, patriarchy and the oppositional capacity of women as agents who could appropriate and negotiate transformative resistance for moral, cultural and political emancipation in a democratic space [10,11]. In addition, there is less of a culture of feministic political struggle in STEM education for equality awareness and empowerment. This lack of equality awareness and politics of emancipation has led to the continued devaluing of females in education and society. This challenge is traceable to the lack of in-depth analysis of the reproduction of cultural stereotyping, epistemic injustice or exclusion from political, social-equality knowledge, in science education and the social space [12]. On the global scene, in Africa and in particular Nigeria, the persistent undermining and demeaning paradigm excluding females' collective political and intellectual struggle position us to go beyond superficial interrogation of women's oppression in the science education space. This is vital because the oppressive gender practices have stifled female pre-service teachers' interest, conceptual understanding, creativity, intellectual capacity, motivation to study science and career progress [13,14]. The demeaning space of subordination and marginalization is disturbing because inequality linked to stereotype threats affects emotional stability, critical thinking, conceptual understanding and cognitive processes by slowing down memory capacity to function during democratic, multicultural and inclusive learning engagements [15,16]. Furthermore, female pre-service teachers suffer even after graduation with greater consequences as women in their workplaces and also in their cultural settings [17,18].

In the following section, three key concepts in gender studies, essential to unravelling the phenomenon of gender inequity and discrimination, are discussed—cultural production, reproduction and subversion. Cultural production is the cultural gender practices that are normalized or legitimized for males and females in society. Reproduction of gender stereotypes is the way cultural discriminatory gender norms and practices are reproduced and reinforced in science, technology, engineering and mathematics education and in particular science education [19,20]. Subversion is perceived as the way that largely females and girls, but not excluding males, appropriate transformative resistance or opposition through the use of gender, power and cultural ideology to reject school ideology that threatens their humanity and dignity [21]. The triad of gender, power and patriarchy could be viewed as forces that complicate the cultural production and reproduction of gender stereotypes globally and likely in Life and Physical Sciences in the college of education

in Nigeria [22,23]. Therefore, when active critical pedagogy is adopted, it could challenge educators' and students' identities to be critical of the language of inequality, as in words, actions, expectations and stereotypes. In this sense, pre-service teachers could be conscious of equality skills and gender pedagogies that have the potential of combating inequity, challenging stereotypes and clarifying misconceptions during class engagements. These equality models could be achieved through inclusive and collaborative engagement, critical thinking, equality and quality strategies such as counter-hegemony and organized political activism devoid of fear or coercion [24,25].

Despite the dearth of studies on feminism in teacher education in Nigeria, the nation differs considerably from other western democracies in the delivery of gender equality pedagogies, creative and/or inclusive teaching and learning strategies in science, technology, engineering and mathematics classrooms. Some studies resonate with our argument that there are entrenched gender inequalities in STEM education and leadership positions globally, and in Africa this needs deeper qualitative foci and scrutiny emanating from classroom research [26,27]. Even in developed countries where issues of gender equality are prominent, the discriminatory practices by science teachers are still evident. For example, Nissen and Shemwell [28] (p. 3) using pre-test and post-test surveys, showed how female students in introductory physics modules experience "chilly" physics classrooms when male instructors spend disproportionate amounts of time talking to male students, ignoring female students' questions and undermining their intellectual ability. These subliminal gender practices reinforce inequalities linked to skewed exposure of female science experiences, causing high anxiety and low performance in their physics modules [29]. Reddy also claims that female students in South Africa tend to have higher anxiety than male students in physics modules, due to the fear of assessment outcomes and failure perceived by males [29]. We advocate for a conscious need to mainstream feminist curricula and to be taken seriously by administrators, educators and the government. Our combined experiences of teaching undergraduates and postgraduates and promoting academic professional development provoke renewed commitment to increasing gender equity awareness in science, technology, engineering, and mathematics Higher Education teaching pedagogy. Therefore, we set out to learn through the voices of educators within Life and Physical Sciences spaces by interrogating and trying to understand their personal reflections on gender practices, beliefs and political stances underpinning differential gender engagement. This study might chart a new phase in the global struggle around equality, sexual democracy, critical thinking and gender transformation in STEM and STEAM. Though STEM and Science Technology, Engineering, Arts and Mathematics (STEAM) education are interrelated, and vital scientific frameworks to promote scientific skills, collaborative learning and conceptual understanding of science concepts during class engagement, we are concerned with different conceptions of Arts by researchers and its inclusion in STEM, hence our focus on stem not steam [30]. Though STEAM is vital in promoting students' creativity, emotional arts and cognitive ability, the potential of STEAM to ensure high-quality creativity, learning outcomes and skills is still being debated. The inclusion of arts into stem was subservient to steam, which seems to promote minimally the learning outcome that expands beyond stem content and skills [31,32].

## 2. Background to the Study

It is of great concern that cultural gender discrimination and stereotypes reproduced in Science, Technology, Engineering and Mathematics (STEM) fields discourage females from participating and performing maximally in their science engagements, thus devaluing the contributions of women in science endeavours [10]. Added to the debates are culturally entrenched implicit differentials that reinforce that sciences are difficult subjects [33]. Furthermore, including biased discriminatory discourses, stereotype images in text materials and explicit writing depicting women as stereotypes in writing threaten their collective existence and progress in society, attempting for gender equity. Because of these overt and covert stereotypic practices, marginalized females suffer from oppressive practices

as a result of a stereotypic assault in education and the social world. Thus, females are engulfed with anxiety, low-self efficacy and poor self-concept to study STEM [34,35]. It is disconcerting that science educators' and pre-service teachers' interactions in complex science education spaces are still entangled with dominating and oppressive practices that impede their intellectual, political, moral transformation and career development, impacting negatively on their protégés.

### 3. Problem Statement

This study seeks to problematize the production, reproduction and subversion of gender stereotype beliefs and views in a college in Nigeria, Africa. This could address the discriminatory gender practices that stifle females' participation, interest, motivation and self-efficacy to study science.

It is problematic when entrenched power relations and patriarchal orientations hinder females from progress and attaining equity [23,36–38]. Studies in gender stereotyping reveal inequity through the entrenched production and reproduction of cultural oppressive practices that are also responsible for diminishing interest, underrepresentation, and low motivation of females in physical and life science classes in education. Furthermore, science education space is not gender neutral; it grants more privileges and dominant space to boys and male educators, boosting male students' morale and interest to learn while discouraging girls to participate and contribute effectively to scientific knowledge. Studies also expose the coded dominant capitalist reality that kept women in perpetual bondage of intellectual and economic oppression [23,39]. Although we recognize the shortcomings of the neoliberal academy located and embodied in everyday economic practices and in STEM and STEAM, we all act within as academics and still benefit from it. Gender stereotypic practices are noticeable in education and stifle intellectual engagement, critical attitude and collaborative spirit of science students with profound effects on female pre-service teachers' confidence and performance to study science. Even though policies and curricula enactment are concurrent with international documents, the implementation suffers setbacks in the science classroom due to gender complexity. Further setbacks are the actioned cosmetic changes in these policies, curricula and legal documents. Therefore, there is a need for individual consciousness to reconfigure the social and political spaces and break the circle of cultural reproduction of gender discrimination in education through academic discourse, transformative activism, social media and advocacy. This could be a possibility to reinvent new historical, social, cultural, economic and political unfair norms for emancipation and transformation.

### 4. Research Questions

The research questions addressed are: (i) What is the nature of science educators' production, reproduction, and subversion of gender stereotype beliefs in science classes in Nigeria? and (ii) How do these science educators produce, reproduce, and subvert gender stereotype beliefs in science classes in Nigeria?

### 5. Theoretical Framing

Theoretically, our understanding of gender, equality, and liberation awareness in STEM Higher Education (HE) curriculum and pedagogy draws attention from three core and interrelated insights, such as Critical Theory, Feminist Reproduction Theory and Critical Feminist Pedagogy (CFP), which underpin this study. Thus, we explored the nature of science educators' reproduction and subversion of gender stereotyping and subversion in science classes. Critical theory is positioned to liberate the oppressed and marginalized individuals by raising awareness on how subordination could impact their critical thinking, collaborative learning, humanity and development. Drawing from Critical Theory that seeks to liberate and transform individuals, Feminist Reproduction Theory is a collective political consciousness aimed at interrogating the way schools and/or STEM educational institutions seek to oppress and subjugate women through patriarchy, gender and economic

orientations [23]. Feminist theorists argue that education, specifically STEM and the social-economic world, deliberately frustrate and marginalize females through coded oppressive practices that dehumanize and treat women as others in the social geography [23,40]. The entrenched deceptive appearances of oppression over the years as reinforced currently in text materials and in science education structures has affected and skewed men/boys and women/girls' participation, critical thinking and creativity in society. This is due to the violent space of inequality and discrimination that patriarchal societies have propagated and reproduced over centuries of abuse. This could stifle women's interest, motivation, participation in science research output, intellectual engagements, and economic development. This demeaning space of marginalization has both short- and long-term effects on men's and women's emotional stability, and economic and intellectual capacity in STEM education and society [41].

In transforming women and girls from economic and intellectual oppressive contexts, the concept of counter-hegemony could be rekindled. Counter-hegemony is a collective and political struggle aimed at re-arranging or packaging the science, technology, engineering and mathematics classroom interactions for conceptual understanding, creativity, and the democratic and emotional transformation of teachers and students [23]. It thus enables possibilities for peaceful co-existence and intellectual empowerment through power sharing that could revitalize democratic and critical consciousness [42,43]. Rethinking counter-hegemony, a tenet of feminist reproduction theory in science, technology, engineering and mathematics education would position science educators and pre-service teachers in the colleges of education to use personal power and gender, then transform into a collective political power and for intellectual freedom and economic development. Furthermore, the critical feminist reproduction model could mediate and bring about the action between a normalized version of inequality and lived experiences of females/males' subjugation in the complex social world and in science, technology, engineering and mathematics education, including STEAM. This pedagogic model may likely produce critical insights, reflections, equality actions, and emotional transformation of both individuals and the broader society where dominant epistemological stances continue to marginalize and subjugate individuals. This approach resonates with critical pedagogues' assertion that oppression can only be dismantled if the oppressed are aware of the problem, do not remain silent, and decide to subvert the mechanisms that reinforce inequalities in society to come out of their marginalization and dehumanization [23,44,45]. We align our thought with Morris et al. [1] who argue for a need to expose cultural, historical and economic limitations and address such silences making historical, political, socio-cultural norms and power dynamics more visible in the classroom. In this light, the end point of science, technology, engineering and mathematics education is critical thinking for the transformation of power relations through critical awareness and engagement to reflect on and ultimately change the world for peace and sustainable development [46,47]. Next is the research methodology that was actioned.

## 6. Research Methodology

This study is a qualitative approach with a case study design embedded with multiple units of analysis. Six participants, two from each of the biology, chemistry, and physics departments participated in the research. A qualitative approach, which aimed at interrogating the phenomenon beneath the superficial understanding of the problem, to reveal the nuances of the phenomenon-case reproduction of gender stereotyping and subversion was used [48]. The case study was used to explore the phenomenon under investigation using different data gathering sources such as observation, questionnaires, and interviews. Rigour and credibility were established, especially where boundaries between phenomena of reproduction of gender stereotyping, subversion and contexts of the participants, are complex [49,50]. Purposive sampling using convenient criterion and maximum variation techniques were used to select the six participants out of the eleven who completed and submitted the questionnaire. Permission and consent were sought

from the college administration and the participants, respectively. These were granted. The six participants were also informed that they could withdraw from the research if they wished to do so. Participants' confidentiality to participate in the study was protected. Since the six participants are familiar with each other, and biology, chemistry and physics departments are centrally positioned in the school of science, this made it difficult for anonymization. To increase anonymity, we gave every educator a pseudonym. Interview quotes and descriptions of the educators were anonymous. A questionnaire, an observation schedule, a semi-structured interview and researcher field notes were the instruments used for data collection. These instruments were first pilot tested to ascertain their face, criterion and catalytic validity. This added trustworthiness value to the research process, including a transformative perspective occurring while interviews were conducted. Member checking and triangulation also ensured the trustworthiness of the data collected and analyzed. Eight weeks were spent collecting data from engaging with educators both in and out of their classrooms. Field notes and audiotaped recordings of the six science educators were conducted during the three focus group discussions, comprising two science educators from each discipline.

The selection of the college as the context for the study was the ease of accessibility to the educators and the knowledge about gender issues in science education. During the data generation and analysis, a coding system based on convenience thematic approach to producing nuanced and in-depth data was used. In this sense, the questionnaire administered and returned by the six science educators was analyzed descriptively, due to the small sample size that limited the conduct of inferential statistics. Hence, no hypotheses were formulated nor a test of effects, relationships, and correlations. The six educators were also interviewed and observed in the classrooms. The data from the data sources were analyzed by synthesis, picking, and sorting out the thick and rich data that produced themes. The section to follow discusses the data and analysis.

## 7. Data Presentation and Analysis

In this section, the data are presented based on the questions, interviews, actions observed from lessons and responses of the participants. This was necessary to link up the data with participants' responses for study coherence.

*Data Presentation*

The six science educators came from biology, chemistry and physics departments, comprising two females and four males. We elicited the data from the participants as they responded during the interviews, lesson observations and questionnaire. Some questions and data snippets from the participants are provided herein in this table based on the following questions:

- Do you engage pre-service teachers equally during science engagements?
- Why do female pre-service teachers give activities to male pre-service teachers to assist them with?
- During science teaching do male and female pre-service teachers interact equally?
- How do you challenge stereotypic practices in the science classroom?

In summary, data from questionnaires, interviews and lesson observations reveal discrimination by educators and male students against female pre-service teachers. However, subversive strategies can help educators and male pre-service teachers reduce stereotypic practices in the science classes.

All the data analysis generated from the questionnaire, observations and semi-structured interviews produced the following themes

## 8. Discussion

The discussion is premised on gender production and reproduction of the five themes generated from questionnaire, interviews and lesson observations as evidenced in Tables 1 and 2.

**Table 1.** Shows Data from Questionnaire, Interviews and Lesson Observations.

| Questionnaire | Interviews | Lesson Observations |
|---|---|---|
| No, I interact more with male pre-service teachers because they have more power and are faster than girls to do class activities- Tengo<br>Due to economic beliefs, benefits and history, we discriminate boys and girls in the society and science classes- Nigam | Most of the time I engage boys more than girls without knowing- Dabomi.<br>I treat boys differently from girls because my parents trained me to behave so- Botko | Educators sometimes favor boys during intellectual activities/questioning than females during engagement |
| I give boys activities that involved strength because they are agents and are stronger than female physically not mentally- Botko<br>Male students are socialized in the culture to be strong, leaders, emotionally stable and brave, while females are trained to be quiet, passive and respectful- Tengo | Females sometime ask male pre-service teachers to do their work and perceive male students to be more capable due to culture and patriarchy- Zitma | Observation reveals male dominating class engagement while educator sometimes watches |
| Female students sometimes are shy and passive to engage in intellectual activities such as demonstrations and questioning- Botko<br>Historical, cultural and political socialization process also contribute to gender discriminations- Dabomi | I think females are trained from the society or culture that women are weaker people, so females always need help from male pre-service teachers- Nigam<br>Both staff and science students can help challenge inequality and stereotyping through encouragement and motivation to stop discrimination in the science class- Dabomi | Most male pre-service teachers were active and took headship position than females and some few male pre-service teachers were passive too |
| Gender stereotype could be challenged through using power and advocacy to reduce inequality- Tengo | We can oppose discrimination through political actions and groupings- Botko | Educators sometime do not challenge discriminatory practices of boys in the class |
| Yes, inequality could be resisted by educating boys and girls of the negative effects- Nigam<br>Yes, sometimes I tell girls to sit up, sometimes I forget and remain silent- Zitma | Educators should form associations and groups to challenge gender stereotype to transform students inside and outside the science classes- Zitma | Neither boys nor females' pre-service teachers challenge discrimination amongst themselves. Educator sometimes rebuke or resist males who dominate female pre-service teachers |

**Table 2.** Gendered Cultural Production, Reproduction and Subversion Themes Emanating from the Six Science Educators.

| Production | Reproduction | Subversion |
|---|---|---|
| Cultural norms such as patriarchy and discrimination of females in the society (Religion and culture) | Females' multiple oppressions in the college of education are reproduced due to entrenched socialization in the society | Educators challenging gender discriminations and stereotype in the science classroom |
| Gender performance based on masculine roles such as strong and powerful activities and feminine roles such as weak and subordinate practices in the society | Educators assigning roles to pre-service teachers based on strength and performance of classroom activities | Educators boost female pre-service teachers to learn during class engagement |
|  |  | Unequal distribution of subversive practices by educators lead to complacency or lack of awareness of potential danger of stereotyping |

### 8.1. Gender, Patriarchy, Home, and Power Intersect Students' Engagements

As an example, the interview conducted with Zitma at the college premises is insightful about her upbringing and the way she treated her own children in terms of gender practices.

Interview: How did your parents treat you and your siblings at home?

Zitma: My parents treat both boys and girls equally at home. But I chose to work with boys at farm not to discriminate them. Later, when I got married, I gave birth to five children, 3 boys and 2 girls. In the house I told them that there are no feminine or masculine jobs. So, the boys cook, and their food was more palatable than the females. In fact, the boys push to cook when they are not satisfied with the food cooked by the sisters. Also, both the girls and boys wash and iron my clothes. So, there was no discrimination among the sexes. I encouraged my male children not to discriminate as men practice in the village, but do as girls do, by cooking, washing plates and ironing my clothes. Because, while at school nobody will cook for them just as their father did when he was schooling. Often the male children complained of washing the plates after eating. That, it was meant for women not men, but I insisted that girls and boys should be treated equally.

From the above interview snippet, sometimes, I treat boys and girls equally at home and in the school. We observe that gender and patriarchy are strong predictors of the schooling process, science education engagements and social relationships. This is because gender is culturally and socially institutionalized due to the power and patriarchal orientations of parents, science educators and socialization processes. Thus, more power is invested in men/boys than women/girls in the society and science education environment [51].

Female educators, science educators and students with cultural patriarchal mindsets perpetuate and reinforce their marginalization antics in education and the social space, either for intellectual, political and/or economic gains. Aligning with a consumerist model in education and an ideology for immediate wealth acquisition, we de-emphasize a culture of immediate gratification for students and teachers complicating gender engagement in the classroom [52,53]. The discriminatory practices against females in education are re-echoed by Weiler, thus we strongly surmise that the problem rests ultimately in the "lack of depth in our understanding of these females, the school, the class, patriarchal and gender ideology that is embodied in texts and social relationships of power" which discriminate largely against females [23] (p. 290). The entrenched stereotypic practices may be responsible for Gender Based Violence (GBV) in society today with profound psychological effects on most females. A possible way to reduce these oppressive and patriarchal practices against women in higher education and in the college where the study was conducted are more likely to appropriate transformative resistance and counter-hegemony defence strategies for liberation and emancipation if they stand their ground. For example, females in Africa and globally could reinvent and reconfigure strategies and seek freedom through productive resistance. This opposition is evident when females in education who were subordinated over the years currently positioned themselves in leadership positions for freedom, moral and economic development through their subversive actions. For example, females who were discriminated against over the years negotiated and re-negotiated political leadership to become head of department, deans, deputy provost, and acting provost. The female vice-chancellor of Uncal University re-negotiated the contradictory education and political terrain through resistive practices thus positioning herself in the academia. These protestations are in line with scholars in gender studies who argue that intellectual, political, moral transformation and development could only occur when oppression is challenged in a civic manner for peace, career mobility and collective existence [54,55]. Mattson calls the transformative resistance "femininity cloning," where females from a class, race and other social groups uphold their gender and feminine ideals in a way that will disrupt masculinity ideals that oppress them [56] (p.12). In our view, therefore, the decolonizing forces from critical feminist pedagogy and counter-hegemony if substantially appropriated will conscientize science educators and their pre-service teachers and likely other geographical students of the agenda for critical thinking, freedom and liberation. The critical feminist approach and subversive acts in education have transformative potential that girls' equal engagements in education could be possible if power relations that lead to oppression and violence against women are disrupted for freedom. Employing transformative resistance and counter-hegemony reproduction/critical consciousness strategies

for mobilization and political empowerment would empower educators and pre-service teachers with the resilience capacity to come out of oppression and discrimination [57].

### 8.2. Resistance, Counter-hegemony and Negotiation Pathways to Moral and Political Transformation

In this study and trying to make sense of educators' responses from the questionnaire, interviews and lesson observations to the subversion of gender stereotypes, we perceive resistance as transformative opposition and counter-hegemonic tactics, such as organized political struggles that were invoked by educators and students for positive democratic, political transformation and intellectual engagement. This was possible when female science educators and pre-service teachers in the school structure and science classrooms defy oppression by posing resistance to curriculum ideology that stifle their intellectual, critical reasoning, change and transformation. On the other hand, the female and male science educators opposed subordination mechanisms of skewed male leadership arrangements that sought to occupy deanship, deputy provost and heads of department positions in the college. This was a reality achieved due to a negotiation and re-negotiation stance to regain equity status in what was a previously contradictory college space [58].

In addition, due to awareness, re-awakening and transformation that occurred during the data collection, we found strong evidence to support educators' negotiation and transformation consciousness as we perceive the participant's responses from the survey and interviews. A snippet from the survey and interviews is elaborated.

Survey Question: Do you challenge gender stereotypes in the class? How and why?

Interview question: Can gender stereotypes be challenged in science education class?

Life science teacher, Tengo, said: Yes, by teaching male and female pre-service teachers and respecting them equally in the class. I believe they are all born equal as humans, and all are the same. However, sometimes I inadvertently give the male student's preference to demonstrate and carry out activities in the class, largely due to their strength and confidence to perform these activities. However, we need to create equality awareness so that colleagues' and pre-service teachers could be challenging and resisting inequality inside and outside the science classrooms.

Teacher Tengo's belief resonates well with Weiler's [23] argument that "resistance is not only oppressive beliefs and practices but also a more critical and political work such as organized and conscious collective oppositional actions called counter-hegemony" (p. 290). Therefore, for a large-scale effective gender program, a collective struggle that will disrupt how cultural gender practices are reproduced could be challenged in the college for equity and emancipation, is necessary.

### 8.3. College of Education Is a Contradictory Terrain for Learning Gender Equity and Feminism

Complex college of education spaces has made gender equity engagements difficult due to both the power of curriculum ideology, gender, and patriarchal forces combined to induce educators to either train or domesticate pre-service teachers to the existing societal norms rather than educate them for critical skills, conceptual understanding, and liberation. Therefore, when male and female pre-service teachers are domesticated in a complex schooling environment, critical skills, citizenship ideals and moral consciousness would be deprived of liberation and transformation [59,60]. The consequence of oppression is often violence, assault and crises in education permeating their broader community as well. As Paulo Freire, a critical pedagogue argues, "Is education for domestication or liberation"? [60] (p. 5). Further, he avers that in domesticated classrooms, the teacher has all the power, skills and knowledge and undemocratically transfers to students, whereas the students remain passive only to receive anything the educator gives in the complex class. Thus, in this learning space, both the uncritical educators and students are more conscious of grades and certification rather than critical insights, creativity and citizenship for peaceful co-existence and transformation of the world. This kind of undemocratic education is oppressive and suppresses students' critical growth and development for liberation and

morality [19,61,62]. On the contrary, education for liberation aims at democratically sharing power with students that is, boys and girls, for critical reasoning, creativity and intellectual engagement in a just environment. In the liberatory education model, the educator sometimes learns from the students and the students also learn from the teacher through critical, democratic, and collaborative engagements. Further, in an unrestricted model of education, the educator teaches to learn, and students learn to teach the educator. Thus, meaningful and respectful dialogue is evoked, leading towards liberation and transformation of the individual and possibly in the school and cultural environment [23,61,63]. Yet quite worrisome, though the democratic space is important for science engagement, we discovered that in the science classes, the female pre-service teachers used the democratic spaces as a commodity for economic upliftment, conforming the neoliberal model for wealth, ignoring learning, conceptual understanding and moral transformation by defying college rules and regulations that sanction little children during class engagement. We also perceive female pre-service teachers bringing children to the class as an act of resilience and transformed insight to learn due to the economic oppression they suffered over the years [64]. Furthermore, the male pre-service teachers used their power of sexuality and patriarchy to advance relationships in the science class. An example of sexual laissez-faire attitude was evident in the class as revealed by a female science educator during an interview session:

Researcher: What would you say about how male pre-service teachers interact or relate with females in the class?

Teacher Tengo: Well, during lectures they sit together to learn in the life science class. But this day when I was teaching, I noticed some noise at the back and when I focused properly, I was so confused, surprised and angry. A boy was busy dominating and oppressing a girl by distracting her attention from science engagement. The girl using her discretionary and individual power was silent in subverting the male pre-service teacher's actions or conscious stereotypic practice. I then asked the female student to come and sit in front of the class and to see me after the class for advice. We perceive subversive acts and control mechanisms by the educator in the classroom even though the female pre-service teachers were vulnerable in the science classroom and school environment due to masculine dominant practices.

The oppressive actions of male pre-service teachers on females, and girls not challenging male pre-service teachers' actions could deprive them of intellectual and moral emancipation and could be referred to as self-stereotyping or self-endangering. The acts have the potential of marginalizing and/or oppressing pre-service teachers, specifically females from critical academic empowerment, significant transformation and identity development due to unconscious oppression [65,66]. This resonates with Freire's assertion that education and specifically science education is complex and a contradictory space for teaching due to "school ideology and structures" [67] (p. 5). In his prophetic vision that promised hope, he adds that within these contradictions there are possibilities for science educators and students to negotiate, and re-negotiate different oppressive tactics for political, moral, and intellectual transformation. This was evidently shown where the female science educator in the life science class had begun to unpack gender issues—she challenged and rebuked the male pre-service teachers' stereotypic practices to motivate and encourage females in the college [62,68].

### 8.4. Females' Multiple Oppressions Can Be Challenged for Liberation through Activism

Even though there is awareness of inequality, oppression and freedom in education, the consciousness is not adequate to effect change globally and in Africa due to the prevalence of deep-rooted cultural stereotypic socio-political and patriarchal norms. For instance, data from observations, interviews and questionnaires reveals that female pre-service teachers suffer multiple oppression from the college staff, fellow students and its gender-biased organizational structure. Although the female educator suffers from the economic privileges, patriarchy, and political dominance due to the explicit power of masculinity, the female undergraduate pre-service teachers are marginalized, dominated, and some-

times side-lined from critical thinking, intellectual actions and active participation in the science classroom. Further, the education space and the cultural environments sometimes implicitly stereotype female pre-service teachers by re-echoing those physical sciences as difficult for females. We are all too well familiar with science and its discourses that are historically masculine oriented, which mostly convey masculine images [69,70]. Studies confirm the stubborn patriarchal and gender cultural stance of boys and science educators, but this could be challenged through advocacy and activism by the females when their consciousness is raised to subvert gender oppression through counter-hegemony, critical thinking and political tactics [42,55]. Valerie and Parmar call these oppressive structures "triple oppression" [71] (p. 4). We perceive the problematic complex spaces of discrimination as contradictory, multiple dimensional consciousnesses and thought suppressing due to implicit and explicit multiple stereotypic threats females experience. Though women experience multiple subjugations, they sometimes allow and contribute to their own gender stereotyping and oppression woes. For example, they willfully harm themselves by not accepting roles that are perceived and coded to be male-dominated activities in education and their society. In Africa and internationally, most societies are gender biased and equip mostly men to be agentic and more capable, and reinforce the female's entrenched stereotype as others, weaker vessels, and kind individuals [72]. Furthermore, in this study, we perceived another stereotypic model, where both male and female science educators and some female pre-service teachers unequally distributed cognitive and practical activities that deliberately reduce female participation and interest for transformation [20,73]. Their collusion with boys in the science classes to oppress, dominate and frustrate the female pre-service teachers further is evident in the following interview snippet.

Researcher: How do you treat male and female pre-service teachers in your physics class?

Teacher Botko: I treat the females the same way as I treat boys. But sometimes, the female pre-service teachers do not respond to questions and engagement like boys do and I perceived them to be reluctant in the class. So, I use the boys most of the time because they are active and respond faster to classroom engagements. However, I do know I am discriminating but to finish course work I need to use the male pre-service teachers.

Despite the multiple stereotypic practices and contradictions, both educators and pre-service teachers can regain their dignity in gender equity through re-negotiation and transformative experiences through empowerment, such as accelerated economic growth and promoting peaceful co-existence. This resonates with researchers' argument that females face diverse discriminations in the social space, economy and education linked to different roles and lack of transformative experiences and if addressed decisively, could change the world positively [74,75]. Therefore, a global and national advocacy to ensure equity in education and the social world enhanced curricula and policies enactment towards equality. Yet, critical feminist thinking slightly deviated from gender programs on equity but argues that these policy documents were only partial in promoting equal engagement with little attention to gender power relations, patriarchy and females' empowerment. This, we perceived, does not appear enough in educational discourses in Africa and the social world, which need academic reconfigurations.

In Nigeria, the national policy on education encourages all education sectors to embrace equality and citizenship education for upward social mobility. Neither educators nor pre-service teachers are given the conducive space to interrogate the knowledge system that oppresses them. Furthermore, there is limited awareness of gender equity in secondary and higher education and the school science curriculum has little gender equity consciousness that can position teachers and pre-service teachers for critical skills, social justice and a peaceful society [76,77]. This we perceive as a dangerous and treacherous path to achieving gender equality, creativity, social justice and transformation for peaceful and social cohesion in Africa and the global social space. It is important that we rethink Next Generation Science Standards (NGSS), due to its focus on learning by doing and discovery toward individual and collective empowerment and transformation through power-sharing epistemic freedom in the science classroom.

Of course, Next Generation Science Standards (NGSS) encourages transformative science and aims not only to increase perception, but to engage in science by doing, discovering, valuing, and making sense of the social world [78,79]. However, our concern is that although NGSS discourse is significant in providing insight that science is learned by doing, it is limited by the lack of interrogating power relations, political struggle and gender consciousness that could impact learning and position students as agents of change and transformation. Our views resonate with researchers who caution that:

NGSS is not a silver bullet for the optimal transformation of the science classes. However, see it as another reform document designed to suggest opportunities for students to actively engage in knowledge construction themselves—to be doers of science, rather than receivers of facts. In this model, there are inherent contradictions, and students will still mimic practices others have selected as important to learn, and contents others have selected as the foundation to study. Unless students are encouraged towards epistemic agency where they are seen as active agents in knowledge production, we are likely going to position students as receivers of science education facts and practices, even as classrooms adopt NGSS that is perceived as a loose consensus, [80] (p. 2).

We perceive NGSS perspective though significant theorizing knowledge, ignored students' agentic potentials, critical thinking, power for transformation and emancipation through self-discovering, self-reflection, and democratic science engagements. To attain gender-active participation, we suggest that all students, specifically females, acquire critical skills and political strategies, and co-produce meaningful knowledge that would be empowering for the college and their community [37,38].

*8.5. Gender Is Linked to Performance Roles and Identity in Science Education*

Gender relations are found to dictate what goes on in the socio-cultural and college structure linked to different performance and identity construction of female and male pre-service teachers. The gender–cultural ideological precepts, such as dominant roles empowered by men and particularly male pre-service teachers as leaders, heads and superior beings, infiltrated the schooling process and the college to determine identity formation and role performance in science education [54,81]. Nevertheless, men are accorded preference and paid higher salaries than females who hitherto were moral and community buffers now occupying leadership positions. Although most gender discourses revolved around the macro level rather than the micro level to highlight entrenched discriminatory practices, we focused on the micro level to explore the cultural reproduction of gender stereotypes to unravel perceived and coded power relations and patriarchy underpinning the oppression of females. Therefore, with this deceptive consciousness, we discover that science educators perpetrate and reinforce culturally explicit stereotypes in the form of dominance and unequal treatment of male and female pre-service teachers where class engagement is skewed against females. Thus, male and female pre-service teachers are socialized into different activities linked to masculinity and femininity identities and ignore the equality positioning of the national policy on education for all Nigerian students and the citizenry [76]. Butler constructed these skewed gender roles and identity construction differentials as "gender performativity" due to continuous act and identity formation [39] (p. 5). A negative spinoff is the likely result of females' low self-efficacy, low motivation and lack of confidence to perform and pursue science agenda due to continual stereotypic practices [82]. We advocate for students to organize learning and teach themselves, because of power and ability, creative skills inherent in them to learn collaboratively and democratically in education. Therefore, the presence of demeaning spaces of gender oppressive orientations cannot mediate between transforming roles and performance of females due to their low performance and meritocracy drive, particularly in science education. The ensuing result is underperformance and eagerness to earn or acquire only certification rather than attain a 'meritocratic consciousness' [37,61]. We also perceive that with entrenched neoliberal ideals of certification as part of the global capitalist agenda, the pre-service teachers sometimes induce educators' test and assessment scores using their sexuality

as a form of commodification and commercialization for material gains, ignoring critical skills, morals, and collaborative and intellectual possibilities [37]. This 'commodification of sexuality' was evident during an interview with female teacher Dabomi.

Female Teacher, Dabomi: I was shocked to my bone marrow when a female pre-service teacher missed my test. She came to my office and exposed her sexual parts wearing transparent clothes to seduce me to give her a makeup test. I was upset and angry. In the next class, I told the entire class that no female should come to my office for any thing again. But within few days I realized that I was too harsh on the females and my action could affect their physics engagements and academic endeavor. I receded on my decision by informing the class that the office was opened to all students regardless of gender. This act was to encourage both genders to learn despite her drive towards a makeup test, certification and probably sexual abuse.

Bourdieu and Passeron [83] argue that merit-based rewards are a function of the capitalist system of stratification to perpetuate and reinforce oppression and stereotypic practices in society and the world that relegate females to the subordination class. In our views, the examination score and certification acquired through material inducement might defile or ignore critical thinking, science skills, creativity and gender citizenship experience for upward intellectual mobility and empowerment. Figure 1 is a proposed Pedagogic Transformative Model based on Critical Feminist Pedagogy that can drive intellectual and political emancipation during science engagements.

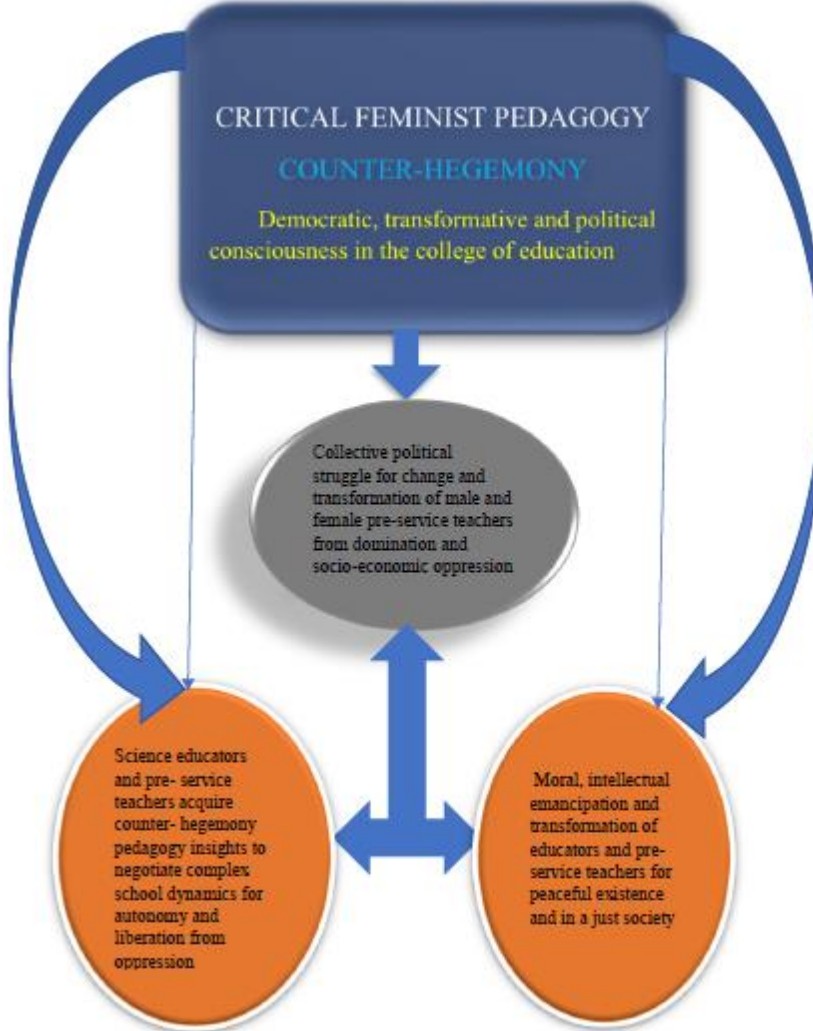

**Figure 1.** Pedagogic Transformative Model based on Critical Feminist Pedagogy.

This transformative pedagogy, if internalized, will position educators and pre-service teachers to subvert stereotypic practices in the science classroom for career growth, moral and political empowerment.

## 9. Conclusions

The presence of discriminatory gender practices in higher education should be used as an opportunity to effect permanent change and to provoke insights to challenge the male-dominated canon for conceptual understanding, change, emancipation, and positive transformation. This study has revealed the nature of explicit and implicit cultural reproduction of gender stereotype beliefs and subversion practices in a college as a microcosm of African/Nigerian society that is linked to cultural norms and dominant power relations. We established that educators bring to the science classroom assumptions and beliefs about the powerful masculine image of males as strong, powerful, emotionally neutral and dominant leaders, and bring the weaker feminine image of female pre-service teachers as passive, illogical, emotional and subordinates due to a lack of gender equity awareness. Educators and male pre-service teachers reproduced skewed stereotypic practices against female pre-service teachers because of entrenched cultural, historical, socio-political power differentials and current patriarchy embedded in their individual and collective sub-consciousness which conflate to drive complex classroom engagements. Undoubtedly, within the contradictions in science education, there are possibilities for negotiation and re-negotiation of the school structure, the curricula ideology for attaining gender equity, critical skills, collaborative action and transformation. This can be made possible through organized emancipatory resistance, feminist counter-hegemony, and rethinking activism/advocacy. In addition, it is possible that the introduction of Critical Feminist Pedagogy in the curriculum is the antidote to oppression and the pathway to promote subversion, enhance equality, and transformation for 21st century gender consciousness.

**Author Contributions:** Conceptualization, D.A. and N.G.; Methodology, D.A., N.G. and A.J.; Validation, D.A., N.G and A.J. All authors have read and agreed to the published version of the manuscript.

**Funding:** This research received no external funding.

**Institutional Review Board Statement:** Ethical certificate with full approval and reference number HSS/1028/0180 was obtained before the study was conducted.

**Informed Consent Statement:** Informed consent was obtained from all subjects involved in the study.

**Data Availability Statement:** Data can be found at School of Education, University of Kwazulu -Natal, South Africa.

**Conflicts of Interest:** The authors declare no conflict of interest.

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
