# Peer review of "Cultural Production, Reproduction and Subversion of Gender Stereotyping among Pre-Service Science Teachers: Insights from Science Educators"

_education, doi:10.3390/educsci12090621_

Round 1
Reviewer 1 Report
The topic discussed in the article is interesting and could contribute to a transcendental field of study in science teaching in general.
However, the study sample is very limited (6 individuals). A more in-depth and well-grounded analysis of the case study would be required for the research results to be robust.
In relation to the methodology used, the researchers use varied methods from which they could extract numerous data, however, the document presents few results that do not support the conclusions of the work.
On the other hand, the answers to some of the questions in the questionnaire are presented in the "Discussion" section, which should be in the "results" section.
Finally, the complete questionnaire does not appear, nor the answers of all the individuals to the questions of the questionnaire.
My recommendation is that the authors restructure the article in depth and add all the information collected in the results so that the scope of the work can be understood.
Author Response
Point 1 The study was a qualitative case study with questionnaire and lesson observations complimenting qualitative study which is the major phase. However, we revised the analysis section suggested
Point 2 More results added to support the conclusion of the work (see table 1).
Point 3 Answers to some questionnaire questions are presented in the discussion section for coherence of ideas, but more answers are captured in result sections now (see table 1).
Point 4 A copy of the questionnaire is attached and individual answers to the questions in the questionnaire reflected (see table 1)
Point 5- We restructured the article and added information collected in the results to enhance scope and understanding of the work.
Reviewer 2 Report
The work you propose is interesting, complete, and the topic is of great interest. However, there are a number of aspects that need to be improved, which I will discuss below.
- In the introduction and theoretical framework, you need to revise the STEM concept. If it is possible you need to introduce more recent research. Why not talk about STEAM instead of STEM?
- There are doble spaces along the manuscript, they should be deleted (on lines 76, 85, 88, 100, 103....)
- In section "data presentation and analysis" you need to improve the presentation of the questionnaire. I think it would be better to use a table or figure to show the data collection tool more visible.
- The title of tables and figures should be better marked.
- The discussion should highlight more when quoting a question from the questionnaire (for example in italics).
- Figure 1 has a disproportionate size in relation to the text, it does not respect the margins of the manuscript.
Author Response
Response to reviewer's comments
Point 1 Recent stem concepts such as critical thinking, collaborative learning emotional stability and conceptual understanding are added to all sections.
Point 2 Double spaces on lines 76, 85, 88, 101, and 103 deleted.
Point 3 Collection tools questionnaire, interviews and lesson observations and data visibly reflected on table 1
Point 4 Titles and tables marked.
Point 5 Questionnaire reflected in italics in discussion section
Point 6 Figure 1 reduced to respect margins of the manuscript
Reviewer 3 Report
Extensive editing of English language and style required
Author Response
Point 1 Editing of English language and style is done
Round 2
Reviewer 1 Report
Congratulations for the work developed.
Author Response
Responses to reviewer's comments
1 Stem and Steam concepts conceptual understanding, collaboration, creativity are added to introduction and theoretical framing.
2. Pages 76, 85, 88, 101 & 103 double spaces are removed
3. Data collection tools and data reflected on table 1 pp316-319
4. Titles and tables are marked in italics pp 316
5 Questionnaire/ educators responses are reflected in italics in discussion section pp339
6. Figure 1 reduced to respect manuscript margin p637
7 Design (Case study) and Research Questions are stated p262
8. Descriptive questionnaire was used not inferential statistics hence hypothesis were not stated.
9. Extensive language editing and spell checks done by language editor. See manuscript with track changes.
